# Anti-Virulence Potential of a Chionodracine-Derived Peptide against Multidrug-Resistant *Pseudomonas aeruginosa* Clinical Isolates from Cystic Fibrosis Patients

**DOI:** 10.3390/ijms232113494

**Published:** 2022-11-04

**Authors:** Marco Artini, Esther Imperlini, Francesco Buonocore, Michela Relucenti, Fernando Porcelli, Orlando Donfrancesco, Vanessa Tuccio Guarna Assanti, Ersilia Vita Fiscarelli, Rosanna Papa, Laura Selan

**Affiliations:** 1Department of Public Health and Infectious Diseases, Sapienza University, p.le Aldo Moro 5, 00185 Rome, Italy; 2Department for Innovation in Biological, Agro-Food and Forest Systems, University of Tuscia, 01100 Viterbo, Italy; 3Department of Anatomy, Histology, Forensic Medicine and Orthopedics, Sapienza University of Rome, Via Alfonso Borelli 50, 00161 Rome, Italy; 4Research Unit of Diagnostical and Management Innovations, Children’s Hospital and Institute Research Bambino Gesù, 00165 Rome, Italy

**Keywords:** antimicrobial peptide, biofilm, invasion, *Pseudomonas aeruginosa*, anti-virulence, protease, cystic fibrosis, cytotoxicity

## Abstract

*Pseudomonas aeruginosa* is an opportunistic pathogen causing several chronic infections resistant to currently available antibiotics. Its pathogenicity is related to the production of different virulence factors such as biofilm and protease secretion. *Pseudomonas* communities can persist in biofilms that protect bacterial cells from antibiotics. Hence, there is a need for innovative approaches that are able to counteract these virulence factors, which play a pivotal role, especially in chronic infections. In this context, antimicrobial peptides are emerging drugs showing a broad spectrum of antibacterial activity. Here, we tested the anti-virulence activity of a chionodracine-derived peptide (KHS-Cnd) on five *P. aeruginosa* clinical isolates from cystic fibrosis patients. We demonstrated that KHS-Cnd impaired biofilm development and caused biofilm disaggregation without affecting bacterial viability in nearly all of the tested strains. Ultrastructural morphological analysis showed that the effect of KHS-Cnd on biofilm could be related to a different compactness of the matrix. KHS-Cnd was also able to reduce adhesion to pulmonary cell lines and to impair the invasion of host cells by *P. aeruginosa*. A cytotoxic effect of KHS-Cnd was observed only at the highest tested concentration. This study highlights the potential of KHS-Cnd as an anti-biofilm and anti-virulence molecule against *P. aeruginosa* clinical strains.

## 1. Introduction

The emergence of multidrug-resistant (MDR) bacteria is responsible for about 15.5% of hospital acquired infections with about 0.7 million deaths from drug-resistant diseases [1]. The Infectious Diseases Society of America focused attention on a fraction of antibiotic-resistant bacteria, enclosed in the acronym ESKAPE (*Enterococcus faecium*, *Staphylococcus aureus*, *Klebsiella pneumoniae*, *Acinetobacter baumannii*, *Pseudomonas aeruginosa*, and *Enterobacter* spp.), so called as they are capable of ‘escaping’ the biocidal action of antibiotics and evading the immune response of the host [2] due to the acquisition of antimicrobial resistance genes; the result is a reduction in the treatment options, especially for chronic and severe infections. This increases the burden of disease and mortality due to treatment failure. Taking into account these considerations, it is desirable to set up a coordinated global response for antimicrobial resistance surveillance [3].

Within the ESKAPE pathogens, the versatile opportunistic pathogen *Pseudomonas aeruginosa*, responsible for both acute and chronic infections, is included. It causes health care-associated infections correlated with high morbidity and mortality in individuals affected by pneumonia, chronic obstructive pulmonary disease (COPD), or cystic fibrosis (CF) [4]. The genetic disease CF is caused by a variety of mutations in the gene coding for the cystic fibrosis transmembrane conductance regulator (CFTR) protein responsible for ion transport across the cellular membrane. Due to an alteration in ion flow, this leads to the production of thick, sticky mucus that clogs the airways, trapping the microorganisms [5].

Therefore, elevated osmotic stress and high concentrations of antimicrobial drugs, together with reduced nutrient availability and interspecies competition, render the lung environment extremely altered. Altogether, these local conditions force *P. aeruginosa* to adopt alternative strategies to escape the host immune response [6]. Indeed, this microorganism possesses a wide and variable collection of virulence factors and antibiotic resistance determinants that enable it to adapt to multiple conditions and survive in hostile environments by producing a specific virulence program [7]. Consequently, *P. aeruginosa* causes highly complex chronic infections.

Among its various virulence factors, the ability to produce highly structured biofilms confers to the bacterium important advantages including phenotypic resistance to endogenous and exogenous stress [8]. Hence, the ability to form a biofilm plays a pivotal role in the colonization of CF patients’ airways by *P. aeruginosa* [9,10].

Biofilm is a dynamic process that allows planktonic bacteria to protect themselves and resist against antimicrobials and evade the host immune response. In a biofilm, bacterial communities are incorporated in a self-produced exopolysaccharide matrix enriched by proteins and eDNA, which ensures their survival in adverse conditions, complicating bacterial eradication [11].

A novel promising strategy for developing new antimicrobial agents aims to inhibit virulence rather than bacterial viability [12,13,14,15]. These compounds, acting on specific bacterial virulence determinants, should not induce the development of resistance, frequently observed for antibiotics [16]. The anti-virulence strategy is particularly suitable for chronic infections related to MDR bacteria such as those responsible for recurrent lung infections associated with CF disease.

In the search for new therapeutic strategies against MDR pathogens, antimicrobial peptides (AMPs) have been identified as new potential drugs to replace or integrate classical antibiotics by overcoming resistance [17].

Different species produce AMPs that contribute to the innate immune response for preventing and/or counteracting pathogen infections; therefore, they represent a ubiquitous class of small proteins (typically 10–100 amino acids long in their biologically active form) usually showing a positive net charge [18].

For this reason, AMPs are promising candidates and emerging therapeutics since they exhibit: (i) a rapid and broad spectrum of antibacterial, antiviral, antifungal, and antiprotozoal activities; (ii) a low risk of developing antimicrobial resistance; and (iii) a low toxicity for the host. In addition, recent evidence has shown the ability of some antipseudomonal AMPs to inhibit biofilm formation [19].

Therefore, in this study, we tested the efficacy of a chionodracine-derived peptide (KHS-Cnd) against *P. aeruginosa* clinical isolates from CF patients. Lots of new peptides have been isolated from different sources in the last few years and, therefore, in order to identify different protein scaffolds, we decided to focus our work on Antarctic marine organisms and, in particular, on the icefish *Chionodraco hamatus*. These animals live in a very extreme environment with a water temperature that is close to 0 °C in winter; they have evolved specific physiological adaptation strategies including antifreeze molecules and specific immunoglobulins. We found an AMP produced by this species, named chionodracine, which is quite different in its mature sequence from other known peptides identified in fish evolutionary related species [20]. This natural peptide was highly active against Antarctic psychrophilic bacterial strains and showed no hemolytic effect against human erythrocytes up to a concentration of 50 µM, but it was not able to kill human bacterial pathogens. Different mutants from the original peptide were successfully designed based on its scaffold to improve the antibacterial activity [18] and especially one, KHS-Cnd, showed a high capacity to kill ESKAPE pathogens including *P. aeruginosa* [21,22].

We investigated the influence of this peptide on different virulence factors such as biofilm formation and protease production, moreover, we determined its cytotoxic effect against alveolar epithelial cell lines. Our results highlight the effect of AMPs on mitigating bacterial virulence, thus representing an important step toward a possible way to control infection and to develop new therapeutic treatments.

## 2. Results

### 2.1. Effect of KHS-Cnd on Biofilm Formation

Phenotypic features of the clinical strains and PA14 reference strain are summarized in Table 1.

Preliminary experiments were performed to assess the effect of KHS-Cnd on planktonic growth of six *P. aeruginosa* strains, five clinical strains, and the hyperbiofilm producer PA14 reference strain [23]. The phenotypic characterization of clinical and PA14 strains are summarized in Table 1. The detected results showed that the peptide did not affect bacterial viability up to a concentration of 40 µM, except for the 32P strain where an antimicrobial activity was evidenced at the highest used concentration. Considering these results, biofilm experiments were performed by testing KHS-Cnd at a concentration of 40 µM for all strains, and only for 32P at a 20 µM concentration.

The effect of KHS-Cnd was investigated either during biofilm development by adding it to the medium at the beginning of the cultivation (time zero, pre-adhesion period), or after biofilm formation (24 h of bacterial culture, mature biofilm). As the control, bacteria were cultured in BHI medium without the peptide. Results of KHS-Cnd on the pre-adhesion period are presented in Figure 1, panel A. Results were expressed as the percentage of biofilm formed in the presence of KHS-Cnd compared to the bacteria grown only in the BHI medium. KHS-Cnd showed an anti-biofilm activity in five out of six tested strains, with a percentage of inhibition ranging between 60% and 20%. The strongest inhibition (60%) was achieved on the isolate 32P and on the reference strain PA14. On clinical strain 40P, instead, a significant increase in the biofilm formation was highlighted. Furthermore, it is worth noting that 40P was the strain exhibiting a lower biofilm biomass at 24 h compared to the others (Table 1).

KHS-Cnd was also tested on the mature biofilm (24 h of bacterial culture). In this case, bacterial strains were grown in BHI medium in microtiter plates for 24 h at 37 °C; then, the growth medium was replaced with BHI containing or not containing KHS-Cnd (40 µM for all strains except for 32P, where we used 20 µM). The microtiter plates were then incubated for a further 24 h at 37 °C. Surprisingly, after the addition of the peptide, in four out of the six tested strains, biofilm reduction was observed, ranging from about 20 to 33% (Figure 1, panel B). Nevertheless, it is worth noting that the biofilm measured after 48 h of incubation was more abundant and doubtless more structured and difficult to eradicate. Interestingly, in the 40P clinical isolate, KHS-Cnd was also able to disaggregate about 25% of the preformed biofilm. These results suggest that KHS-Cnd action is not restricted to the initial bacterial attachment on the abiotic surface, but is also effective on mature biofilm.

### 2.2. Ultrastructural Morphology of P. aeruginosa PA14 and 27P Strains Following the Treatment with KHS-Cnd

Ultrastructural morphology of *P. aeruginosa* biofilm was observed using a variable pressure scanning electron microscope (VP-SEM). This analysis was performed on the PA14 reference strain and the 27P clinical strain treated or not-treated with KHS-Cnd. 27P was chosen because it was a good biofilm former and KHS-Cnd strongly disaggregated its mature biofilm.

The observation using VP-SEM of the untreated *P. aeruginosa* PA14 samples (Figure 2A,B) showed that the sample was characterized by the presence of an abundant extracellular matrix (ECM), whose surface appeared irregular, being smooth and compact (A, asterisk) in some areas while in others, it appeared spongy and rough (A, star). At higher magnification (B), the ECM area with the spongy appearance was appreciable in detail. The three-dimensional network structure was well visible and included larger meshes in some areas (arrow) and smaller ones in others (dotted oval). Network meshes were lined by ECM trabeculae that interconnected one another, developing a regularly branching trabecular system, and the overlapping of the meshes created a labyrinthic system of narrow channels.

The treatment with KHS-Cnd induced relevant ultrastructural modifications, as illustrated in Figure 2C,D. The picture at low magnification (C) showed a spongy appearance of the sample, almost on all of its surface (C, Star); compact and smooth areas were rare (C, asterisk). The image at higher magnification (D) clarified that the spongy areas contained trabeculae formed by aggregates of fine granular filaments, showing a “pearl necklace” aspect (arrow). Comparing Figure 2B of the control sample with Figure 2D of the treated sample, the disaggregating effect of KHS-Cnd was evident. Its action was exerted homogeneously on the sample surface and was not too aggressive. Filaments forming the trabeculae appeared compactly assembled in the control sample, and, after the treatment, were arranged more loosely, increasing the overall sample porosity. The low aggressiveness of the treatment was also revealed by the observation that, even if we had an increase in porosity, this aspect was not dramatically changed: ECM filaments maintained their fine and regularly granulated aspect, and no filaments had curly free endings, which means that filament disruption was observed.

To further quantitatively analyze the effect of KHS-Cnd on the *P. aeruginosa* PA14 ECM ultrastructure, we used the Hitachi Map 3D Software (v.8.2., Digital surf, Besançon, France). The three-dimensional reconstruction of the samples and the circular area extracted are magnified and shown in false colors in Figure 3A–D for a representative experiment. High elevated areas corresponding to the ECM trabeculae lining the meshes are represented in green. The deep channels of the labyrinthic system are represented in blue.

Comparing the images in false colors of the control sample (Figure 3A,B) with the images of the treated sample (Figure 3C,D), the disaggregating effect of KHS-Cnd was evident: the trabeculae in the control sample presented an arrangement with tighter filaments, and the meshes were tight, while the trabeculae in the treated sample had a loose filament arrangement and the meshes were larger.

Images of untreated *P. aeruginosa* 27P samples in Figure 4A showed the presence of an abundant extracellular matrix (ECM), organized in both compact (asterisk) and spongy (star) areas. The observation at higher magnification (B) of the spongy area showed in more detail that ECM was formed by interconnected laminae, rather than trabeculae (as in PA14) with curly edges (l, laminae). The 3D arrangement of interconnected laminae formed a labyrinthic system of channels (larger than that of PA14), producing an ECM spongy appearance.

Treatment with KHS-Cnd showed a less robust effect on the *P. aeruginosa* 27P ultrastructure. The image at low magnification (C) of the treated biofilm showed a spongy appearance on all of the sample surface, namely, less compact when compared to the control sample (Figure 4A). At higher magnification of the treated biofilm (D), the edges of the laminae formed an ECM 3D network, which appeared curly as in the control sample (comparing Figure 4B of the control sample with Figure 4D of the treated sample), and the porosity did not significantly increase.

To further quantitatively analyze the effect of KHS-Cnd on the ECM ultrastructure of *P. aeruginosa* 27P, we used the Hitachi Map 3D Software (v.8.2., Digital surf, Besançon, France). Results are shown in Figure 5A–D for a representative experiment. The three-dimensional reconstruction of the samples and the circular area extracted were magnified and are represented in false colors in Figure 5A–D. High elevated areas corresponding to the ECM trabeculae lining the meshes are represented in green. The deep channels of the labyrinthic system are represented in blue.

Despite no dramatic morphological modifications being appreciable, when comparing the images in false color, the presence of a less compact matrix (Figure 5C) and more abundant labyrinthic channels (Figure 5D) were evident in the treated sample compared to the control (Figure 5A,B).

Using the Mountains Map software, the values of the projected area of holes and peaks present on the ECM surface in the control versus the treated samples were calculated on 50 extracted areas from the biofilm surface for each sample type. These data were statistically analyzed and the results are shown in Figure 6.

We showed that the treatment with KHS-Cnd induced morphological changes that were statistically significant for the *P. aeruginosa* PA14 strain. With regard to *P. aeruginosa* 27P, conversely, the treatment with KHS-Cnd induced modifications statistically significant only for areas related to the peaks.

### 2.3. Effect of KHS-Cnd on Protease Activity

Since extracellular proteases secreted by *P. aeruginosa* play important roles in infection pathogenesis, proteolytic activity was determined by qualitative and quantitative analyses. These tests were performed on the culture supernatants of *P. aeruginosa* clinical and reference strains treated with 40 µM KHS-Cnd (except for 32P where 20 µM KHS-Cnd was used); results were compared with those of the untreated cultures after 24 h of growth.

For qualitative analysis of proteases secreted by clinical isolates of *P. aeruginosa*, the gelatin-zymography assay was performed after SDS-PAGE separation of the supernatant samples. Gel incubation in a renaturation buffer permitted the digestion of the protease substrate, and after Coomassie staining, a clear band indicative of protease activity was detected for all samples without significant qualitative differences among the KHS-Cnd treated samples and their relative untreated counterparts (Figure 7).

In the quantitative assay, proteolytic activity was detected using azocasein as the protease substrate [24]. After digestion by proteases, the produced protein fragments were quantitatively analyzed by measuring their absorbance, after the removal of the undigested substrate by trichloroacetic acid precipitation. Proteolytic activity was normalized to the amounts per microgram of the total proteins present in the supernatant samples. As reported in Figure 8, the results were expressed as a percentage of residual proteolytic activity of the treated cultures in comparison with that of the corresponding controls. The obtained data indicated that the treatment with KHS-Cnd did not significantly affect the protease activity in three tested strains including the reference one. Conversely, significant alterations were observed in the other three clinical strains; in particular, a partial significant reduction in protease activity was highlighted in the 23P strain; whereas a significant increase was observed in the supernatants of the 27P and 31P strains. The latter showed a higher protease production (64%) in the presence of KHS-Cnd compared to the control (Figure 8).

### 2.4. Effect of KHS-Cnd on Eukaryotic Cell Viability

The biocompatibility of KHS-Cnd was investigated by performing a cytotoxic assay on adenocarcinomic human alveolar basal epithelial cells A549. This cell line was chosen because it represents the main target in *P. aeruginosa* infections of patients affected by CF. The cell viability was detected after 4, 6, and 24 h of incubation with the peptide at different concentrations starting from 40 µM. As shown in Figure 9, toxicity was observed for the highest tested concentration (40 µM); in this condition, the percentages of living cells were 69%, 68%, and 55%, after 4, 6, and 24 h, respectively, compared to the control (untreated cells). Despite this, the cell morphology of A549 showed no evident.

At all other tested peptide concentrations (5–20 μM), instead, there was a slight cytotoxic effect since these values ranged from 75% to 87%.

### 2.5. Effect of KHS-Cnd on Adhesion and Invasion of P. aeruginosa to Eukaryotic Cells

KHS-Cnd action was also evaluated on the *P. aeruginosa* ability to adhere and invade human cells. Since the bacterial strains studied in this paper were isolated from CF patients with chronic airways infection, adenocarcinomic human alveolar basal epithelial cells A549 were used. For this experiment, the strain 27P was chosen because its mature biofilm was clearly influenced by KHS-Cnd. First, bacterial resistance to gentamicin was evaluated. Obtained data showed that strain 27P was sensible to this antibiotic at a concentration of 300 µg/mL. Taking into account the high adhesion ability of *P. aeruginosa* to eukaryotic cells, a multiplicity of infection (MOI) of 1:10 was used. The adhesion and invasion efficiency of the KHS-Cnd treated and untreated bacteria are reported in Table 2. Adhesion was defined by the number of bacteria adhering on A549 cells after 1 h of incubation. Invasion represents the number of internalized bacteria in A549 cells that survived after lysis with gentamicin (1 h incubation on cells plus an additional hour for gentamicin treatment). The adhesion efficiency of *P. aeruginosa* clinical strain 27P corresponded to about 10% of the total used CFU (about 10^5^ bacterial cells). Furthermore, our results showed that the adhesion efficiency of the *P. aeruginosa* 27P strain was drastically affected by KHS-Cnd incubation (about one order of magnitude).

Our data demonstrated that about 1% of total bacteria adhering to A549 cells were able to invade the host cells (about 10^3^ bacteria). Interestingly, the invasion efficiency was also reduced after KHS-Cnd treatment (3.0 × 10^3^ ± 0.2 × 10^3^ vs. 8.06 × 10^3^ ± 0.08 × 10^3^). Moreover, the results were also statistically significative (*p* value = 0.00083).

## 3. Discussion

The global spread of multidrug resistant strains is a major concern for many bacterial species including *P. aeruginosa* as they are highly pathogenic compared to other Gram-negative bacteria and are able to survive and persist in various environments. Therefore, it is not surprising that *P. aeruginosa* can be found alive in the hostile lung environment of CF patients, thus evading the host immune system, leading to morbidity and mortality in CF and immunocompromised patients [25]. Unfortunately, this opportunistic pathogen is resistant to common available antibiotics such as aminoglycosides, quinolones, and β-lactams [17,26]. In addition to known intrinsic and acquired resistance mechanisms, *Pseudomonas* also uses an adaptive antibiotic resistance that includes virulence factors such as biofilm formation, motility, and the secretion of toxins and proteases [27]. In the lungs of infected patients, biofilm acts as a physical barrier formed by a robust extracellular matrix (ECM) encapsulating resilient communities of surface-adhered bacteria resistant to antibiotics; bacteria in biofilm are responsible for chronic recalcitrant infections of CF patients. Hence, there is a need to develop innovative approaches that are able to increase the sensitivity of a pathogen to the available therapies. On the other hand, the development of new antibiotics is very limited and time-consuming, and quite costly from an industrial point of view. In this scenario, to counteract *Pseudomonas* infections, novel therapeutic strategies could also be used to implement the conventional ones. In this regard, it could be promising to target *P. aeruginosa* virulence factors playing a relevant role in chronic infections, thus increasing bacterial susceptibility to known antimicrobials [27]. In this context, antimicrobial peptides (AMPs) are new potential candidates as anti-virulent compounds: they could be more useful as anti-biofilm molecules than conventional antibiotics, since their target is the bacterial membrane and their action leads to cell death, regardless of metabolic state [28]. Moreover, several AMPs (such as defensin, LL-37, IDR-1018, GL13K, NLF20, melittin, magainin II, ranalexin, T9W, cecropin P1, indolicidin, and nisin) have shown antimicrobial activity against *P. aeruginosa* via either the reduction/inhibition of bacterial viability or biofilm formation [26,27,28,29]. Most of these anti-pseudomonal peptides have been isolated from humans or animals [18]. In this study, we focused on a novel AMP derived from the so-called Chionodracine (Cnd) peptide, isolated from Antarctic icefish *Chionodraco hamatus*, which represents an untapped reservoir of biodiversity [20]. Starting from the natural peptide sequence (22 amino acids), Cnd was modified by introducing charged residues; namely histidines and serines were all replaced by lysines, thus obtaining the mutant of interest named KHS-Cnd. We previously demonstrated that KHS-Cnd versus Cnd: (i) selectively disrupts bacterial versus mammalian membranes; (ii) displays a reduced cytotoxic activity against human primary cells and a low hemolytic activity; and (iii) shows a significantly high bactericidal activity against the ESKAPE pathogens [21,22].

Given the above, we tested the anti-virulence effect of KHS-Cnd against *P. aeruginosa* clinical isolates from CF patients. We first analyzed the anti-biofilm activity of KHS-Cnd in the pre-adhesion phase of biofilm growth and on the mature biofilm. Notably, anti-biofilm peptides are biochemically equivalent to AMPs: they share the same amino acid length (12–50), amphipathic properties in terms of basic residues (2–9 residues of K or R), and the amount of hydrophobic residues (about 50% on total) [29]. Different AMPs have been demonstrated to prevent biofilm attachment, starting from the observation of their antimicrobial activity on planktonic bacteria [30], although some do not act on mature and well-structured biofilm. We, herein, demonstrated that KHS-Cnd impaired biofilm development without affecting bacterial viability up to the tested concentration of 40 µM (and 20 µM for 32P strain). Similarly, among the AMPs tested against different clinical isolates of *P. aeruginosa* from CF patients, the ability of the IDR-1018 peptide to prevent biofilm formation has been reported, despite its weak activity against planktonic bacterial forms [31].

Regarding biofilm inhibition by KHS-Cnd, it affected the biofilm development of five out of six tested strains with different efficacy, while only 40P strain biofilm formation was strongly promoted by KHS-Cnd; notably this strain was the lower biofilm producer compared to the others (Table 1). Since biofilm development is a multifactorial event, this result is not entirely unexpected. Particularly interesting is, in contrast, the disaggregation induced by KHS-Cnd on the mature biofilm, since it is notoriously very hard to disrupt an established biofilm. The IDR-1018 peptide also displayed disaggregating activity on the mature biofilm produced by *Pseudomonas* clinical isolates, but the underlying molecular mechanisms still remain to be clarified [31]. In general, it has been hypothesized that AMPs’ mechanism of action targets bacterial species-specific properties such as motility, cell adhesion, and quorum sensing [29]. In particular, for the anti-biofilm broad spectrum activity of IDR-1018, the authors evidenced that this peptide induces the degradation of the second messenger (p)ppGpp nucleotides, which play an important role as signals in biofilm formation and maintenance [32].

Several published studies report AMPs’ ability to interfere with various stages of biofilm formation because they prevent initial bacterial adhesion to surfaces or destroy mature biofilms by detachment and/or killing of biofilm-embedded bacteria. Most AMPs are more effective in inhibiting the early phases of biofilm development rather than in eradicating the mature biofilm. This is probably due to the multiple interactions that AMPs may establish with the main components of the extracellular matrix (DNA, polysaccharides, and proteins) that surround and protect bacteria in the mature biofilm [33].

Moreover, we observed a disruption of the mature biofilm in four out of six bacterial strains (Figure 1, panel B). The disrupting activity was even evidenced on the 40 P strain despite the peptide not showing an inhibitory activity during the pre-adhesion period of the same strain.

Notably, there was no effect on the 23 P strain, whose biofilm at 48 h was clearly lower compared to those produced by other strains. We can speculate that the effect of KHS-Cnd seemed to be more marked for stronger biofilm producers. Concerning the 23 P strain, the biofilm measured at 48 h was much thinner than that observed at 24 h. This phenomenon is not surprising, since biofilm development is the result of a dynamic process characterized by a fluctuation in biofilm amplification and the cell detachment phases, as already reported [34].

Biofilm development is a multi-stage process that requires different actors playing in temporally distinct moments in the different phases. In the early stages, bacterial cells adhere to a surface in a reversible way by cell appendages such as flagella and type IV pili and, successively, they switch to irreversible attachment thanks to the involvement of specific proteins [35,36]. Subsequently, the attached bacteria form microcolonies that spread into a more structured architecture and develop three-dimensional mushroom-like structures that are the hallmark of biofilm maturation [35]. Therefore, it is not surprising that in different phases, the same molecule can act to impair or enhance the biofilm development, as observed for the 40P strain. We can speculate that KHS-Cnd could have an effect, in the case of the 40P strain, on structures critical for the mature biofilm rather than in its pre-adhesion stage.

Regardless, the obtained data did not show a homogeneous response of the tested strains to KHS-Cnd. This latter could be attributed to the high genotypic and phenotypic variability of *P. aeruginosa* strains induced by the environmental conditions in the airway infections of CF patients [37]. Bacterial adaptation and evolution involve, indeed, variations that are also in the composition of extracellular matrix structure (e.g., qualitative and quantitative differences in polysaccharide content); the different structures exposed by the extracellular matrix during biofilm evolution can likely act as a target of the AMPs.

VP-SEM analysis provided interesting information on the action of KHS-Cnd on the PA14 reference and 27P clinical strains.

The morphological analysis highlighted that the effect of KHS-Cnd treatment on the biofilm formed by the PA14 strain, and moderately by the 27P strain, is not only related to a reduction in the biofilm amount, but also to a different compactness of the matrix. Bacteria were still mostly immersed in the matrix, but the matrix seemed more porous, without dramatic changes in the ECM filaments.

Concerning protease activity, another virulence factor that we took into consideration, our results showed a strain-dependent effect of KHS-Cnd in terms of increased and/or decreased secretion in three out of six tested isolates. In the literature, little information is available on the protease secretion by *P. aeruginosa* isolates from CF patients, whose pathogenicity is undoubtedly influenced by this virulence factor [38]. It has been suggested that the production of proteases could be a direct and rapid bacterial defense mechanism to counteract the AMPs [39]. Despite this, the 31P strain displayed a significant increase, of about 60%, in protease secretion, showing an inhibition in the pre-adhesion biofilm of 37%.

KHS-Cnd was also able to reduce the adhesion to a biotic substrate such as pulmonary cell lines (one order of magnitude). Consequently, the invasion of host cells by *P. aeruginosa* was also significantly impaired after treatment (the reduction was higher than 50%).

Disrupting a biofilm without killing cells raises concerns about the further dissemination of a pathogen. A feasible solution to this risk could be the administration of AMPs in synergy with conventional antibiotics to overcome this effect [40]. With regard to the possible selection of resistant strains due to selective pressure exerted by AMPs on bacteria, it is conceivable that AMP exposure may lead to a lower number of resistant mutants compared with small-molecule antibiotics [41]. Indeed, if the peptide acts by weakening bacterial virulence and pathogenicity without affecting its viability, the evolutionary driving-force toward the development of resistance is certainly less strong.

## 4. Materials and Methods

### 4.1. Ethics Approval and Informed Consent

The following research was approved by the ethics committee of the Pediatric Hospital and Institute of Research Bambino Gesù (OPBG) in Rome, Italy (No. 1437_OPBG_2017 of July 2017). This study was conducted with respect to the Declaration of Helsinki as aa statement of ethical principles for medical research involving human subjects. All participants and legal guardians signed an informed consent form that was included in the study.

### 4.2. Bacterial Strains and Growth Conditions

Clinical strains of *P. aeruginosa* were isolated from respiratory specimens from CF patients in the follow-up to OPBG. Some of the phenotypic characteristics of bacterial strains used in this work are summarized in Table 1. Bacteria were grown in Brain Heart Infusion broth (BHI, Oxoid, Basingstoke, UK). Bacterial cells were grown in planktonic condition at 37 °C under orbital shaking (180 rpm), while biofilm formation was performed at 37 °C in static conditions.

### 4.3. Peptide

The peptide (98% purity) KHS-Cnd (WFGKLYRGITKVVKKVKGLLKG) was synthesized by Caslo Aps (Caslo Aps Kongens, Lyngby, Denmark) and characterized as previously described [21]. For all experiments, a concentration of 40 µM of KHS-Cnd was used, except for the 32P strain, where 20 µM of KHS-Cnd was used.

### 4.4. Biofilm Formation

The biofilm content was quantified by the microtiter plate (MTP) biofilm assay [4]. An overnight bacterial suspension was 1:100 diluted into the wells of a sterile 96-well polystyrene flat base plate prefilled with medium in the presence and absence of KHS-Cnd at sub-MIC concentrations. The plates were incubated overnight at 37 °C under static conditions. After incubation, the supernatant containing planktonic cells was gently removed and the plate was washed with double-distilled water. Then, the microtiter plate was patted dry in an inverted position. The staining was performed with 0.1% crystal violet for 15 min at room temperature. The excess of crystal violet was removed; the plate was washed with double-distilled water and thoroughly dried to quantify the biofilm formation. The adherent biofilm was solubilized with 20% (*v*/*v*) glacial acetic acid and 80% (*v*/*v*) ethanol, and spectrophotometrically quantified at 590 nm. Each data point was composed of three independent experiments, each performed in at least three replicates.

### 4.5. Mature Biofilm

Assays on the preformed biofilm were also performed [11]. The wells of a sterile 96-well flat-bottomed polystyrene plate were filled with 100 µL of BHI medium containing 1:100 dilution of overnight bacterial culture. The plates were incubated for 24 h at 37 °C in static condition, then the content of the plates was poured off and the wells were washed to remove the unattached bacteria. A total of 100 μL of fresh BHI containing or not containing KHS-Cnd was added into each well. The plates were incubated for an additional 24 h (48 h in total) at 37 °C. After 24 h, the plates were analyzed as described above. Each data point was composed of three independent experiments, each performed in at least three replicates.

### 4.6. Variable Pressure Scanning Electron Microscope (VP-SEM) Analysis

Samples were washed in 0.1 M phosphate buffer pH 7.4 (PB) and fixed in 2.5% glutaraldehyde in 0.1 M PB buffer pH 7.4 for at least 48 h. Samples were then washed overnight in PB and the day after they were postfixed with OsO_4_ 1.33% for 1 h at room temperature. Samples were washed for 20 min with H_2_O and treated for 30 min with tannic acid 1% in H_2_O. Then, they were washed for 20 min with H_2_O; the excess water was dried carefully with filter paper and the samples were mounted on the specimen holder by carbon film and observed in a Hitachi SU3500 microscope (Hitachi, Tokyo, Japan), at variable pressure conditions of 5 kV and 30 Pa [42,43].

Three-dimensional reconstruction was undertaken by Hitachi Map 3D Software (v.8.2., Digital surf, Besançon, France). A single image reconstruction procedure was used, and a representative area was extracted from the 3D reconstructed image. The surface topography of the extracted area was shown in false colors [44] and used to extract data of the projected area of holes and peaks present on the extracellular matrix (ECM) surface in the control and treated samples. Data (from N = 50 extracted areas for each sample) were then statistically analyzed by the software MedCalc© (MedCalc Software Ltd., Ostend, Belgium; https://www.medcalc.org; accessed on 20 September 2022).

### 4.7. Protease Assay

The total proteolytic activity of *P. aeruginosa* was determined by the azocasein assay [45]. A total of 150 μL of both the KHS-Cnd treated and untreated culture supernatants were added to 500 μL of 0.3% *w*/*v* azocasein (Sigma, St. Louis, MO, USA) in 50 mM Tris–HCl, 0.5 mM CaCl_2_ pH 7.5, and incubated at 37 °C for 30 min. The enzymatic reaction was stopped by adding an equal volume of l0% ice-cold trichloroacetic acid and incubating the reaction at 4 °C for 10 min. After incubation, the insoluble azocasein was removed by centrifugation at 10,000 rpm for 10 min, and the resulting supernatant was measured at OD 400 nm. Each experiment was performed at least in three replicates.

### 4.8. Zymography Assay

Zimography analyses were performed on the KHS-Cnd treated and untreated culture supernatants of *P. aeruginosa* clinical and reference strains. Protein samples (0.75 μg) of unconcentrated culture supernatants were combined with Laemmli sample buffer without a reducing agent nor boiling [45]; then they were separated on a 10% SDS-PAGE gel containing 0.2% gelatin (Sigma–Aldrich, Milan, Italy) with a 4% stacker [46]. Pre-stained molecular mass markers were included (mPAGE Color Protein Standard, Millipore, Milan, Italy) and electrophoretic run was performed until the 23 kDa pre-stained marker band was approximately out of the gel. To renaturate proteins by removing SDS, the gels were incubated with 2.5% Triton X-100 for 1 h, at room temperature. After washing in distilled water, gels were incubated overnight at 37 °C in a development buffer containing 50 mM Tris/HCl pH 8.0, 10 mM CaCl_2_, 1 mM ZnCl_2_, and 150 mM NaCl [45].

Gels were stained for 1 h with 0.5% Coomassie blue R-250 in glacial acetic/methanol/distilled water (1:3:6), and destained in distillated water until the clear bands were visible on a blue background. The resulting zymogram images were acquired by using the ChemiDoc XRS+ System (Biorad, Segrate, Italy). Zymogram experiments were repeated twice.

### 4.9. Eukaryotic Cells

The adenocarcinomic human alveolar basal epithelial cells A549 (ATCC CRM-CCL-185) were cultured in minimal essential medium with Earle’s balanced salt solution high glucose 4.5 g/L (MEM/EBSS), supplemented with 10% fetal calf serum (FCS), 1% glutamine, and 1% penicillin–streptomycin in an atmosphere of 5% CO_2_ at 37 °C. All media were from Euroclone (Milan, Italy). Confluent monolayers were used 24 h after seeding.

### 4.10. Cell Viability

The cytotoxicity of KHS-Cnd on adenocarcinomic human alveolar basal epithelial cells (A549) was assessed using the 3-(4,5-dimethylthiazol-2-yl)-2,5-diphenyl-2H-tetrazolium bromide (MTT) cell proliferation kit (Roche Applied Science, Penzberg, Germany). A549 cells were incubated at 37 °C under 5% CO_2_ for 4, 6, and 24 h after cell inoculation, as described above. A 50 μL volume of MTT working solution was added to each well and the mixture was incubated for an additional 4 h. Purple crystal formazan was observed around cells at 40X magnification under a microscope. The cell medium was carefully removed, and then 100 μL of dimethyl sulfoxide (DMSO) was added to each well to dissolve the formazan. After 15 min of incubation at 37 °C to completely dissolve formazan, the absorbance at 490 nm was measured on a Tecan Infinite 200pro microplate. Cell survival was expressed as a percentage of viable cells in the presence of different KHS-Cnd concentrations (5, 10, 20, and 40 μM) in comparison to the control cells. Control cells are represented by cells grown in the absence of the peptide supplemented with identical volumes of DMSO. The Student’s *t*-test was performed for statistical analysis.

### 4.11. Antibiotic Protection Assay

Semi-confluent monolayers (1.25 × 10^5^ cells/well) of adenocarcinomic human alveolar basal epithelial cells (A549) were cultured in 24-well plates (BD Falcon, Corning, NY, USA) in basal medium containing 10% FCS with 1% antibiotic (penicillin–streptomycin) at 37 °C and 5% CO_2_ to obtain a confluent monolayer after 24 h. One hour before infection, the culture medium was replaced with DMEM medium plus 1% glutamine added with 2% CSF without antibiotics. Strain 27P was cultured overnight in BHI broth at 37 °C at 180 rpm. Bacteria were diluted 1:100 in prewarmed BHI and sub-cultured up to OD 600 = 0.5 at 37 °C in the presence and absence of 40 μM KHS-Cnd. Then, human cells were separately infected with the KHS-Cnd treated or untreated bacterial suspension at a multiplicity of infection (MOI) of about 10 bacteria per cell (MOI 10:1) for 1 h (37 °C in 5% CO_2_).

After 1 h, unbound bacteria were removed from the cell monolayers by two washes with PBS. Then, thee cells were lysed with 0.025% Triton X-100, serially diluted, and plated on MacConkey agar (Oxoid, UK) to count the viable adherent and internalized bacteria.

To determine the amount of internalized bacteria, the cell monolayers were washed with PBS; then 500 μL of fresh medium containing 300 µg/mL of gentamicin was added to each well and incubated in the same conditions for 1 h to kill the extracellular bacteria. The sensitivity of bacteria to gentamicin and the absence of toxicity toward A549 cells was previously verified. After this additional hour, cells were lysed with 0.025% Triton X-100 and the collected supernatants were plated on MacConkey agar (Oxoid, UK), followed by overnight incubation at 37 °C to count the viable intracellular bacteria. Data represent the mean of three independent experiments. Adhesion is expressed as CFU of bacteria that adhered to A549 cells 1 h post-infection at 37 °C. Invasion efficiency is expressed as the CFU of bacteria that were gentamicin resistant 1 h post-infection.

### 4.12. Statistical Analysis of Biological Evaluation

Data were statistically validated using the Student’s *t*-test comparing the experimental data of the treated and untreated samples. The significance of differences between the mean absorbance values was calculated using a two-tailed Student’s *t*-test.

## 5. Conclusions

This study highlights the potential of KHS-Cnd as an anti-biofilm and anti-virulence molecule against *P. aeruginosa* clinical strains obtained from CF patients. For a possible therapeutical application, the anti-virulence activity of KHS-Cnd against the preformed biofilms might be its most interesting feature. However, some limitations should be considered. The anti-biofilm activity of KHS-Cnd seems to act at relatively high concentrations, where a toxicity of about 50% was observed on the A549 cells after 24 h of treatment (Figure 6). Despite this, morphological alterations and hemolytic activity on primary cell lines were not observed for the peptide [21].

Given the importance of identifying new anti-biofilm agents that counteract *P. aeruginosa* related infections, our data look promising, as we found a peptide scaffold that could be used as a starting point for drug development through the design of new peptide mutants with optimized and improved biological performance. In this scenario, derived peptides, particularly shorter forms of native ones, which retain the anti-biofilm properties and display an enhanced activity compared to the original peptide, could be selected; moreover, we can quote some examples of derived peptides that also exert a reduced toxicity on eukaryotic cells [47]. Alternatively, a prodrug strategy to limit cytotoxicity but exploit anti-biofilm activity can be explored. This could be obtained by the conjugation of a drug moiety to AMP (forming AMP prodrug or pro-AMP), with consequent reversible peptide inactivation up to reaching the release site, thus confining the potential cytotoxic in vivo effect to the site of infection. This approach has been successfully applied in vivo to the AMP mellitin, which exhibited high in vitro toxicity [48]. Moreover, new pro-AMPs have been also investigated for their use in the treatment of *P. aeruginosa* infection in CF patients [49]. In addition, we need to consider that AMPs could also be used in synergistic combinations with known antimicrobial agents to counteract antimicrobial resistance; in this case, the dosage of AMP added to the antimicrobial should be relatively low, thus avoiding cytotoxicity.

Aside from the improvement in the biological activity, the set-up of delivery systems using liposomes or nanoparticles (NPs) is fundamental to enhance the bioavailability of peptides within the *P. aeruginosa* biofilm and throughout the thick mucus clogging the tracheobronchial tree of patients with CF or COPD. In this challenging context, the use of robust pro-AMPs or AMPs conjugated with nanocarriers represents a resolutive tool to improve peptide stability and selective targeting; this strategy reduces both toxicity and the overall dose, and allows for a controlled local release of the AMP. Once the pharmacodynamic behavior of the AMP, encapsulated within NPs, has been deeply investigated, dosing regimens can be rationally optimized: this aspect also needs to be addressed to guarantee a clinical impact of projects focused on AMPs, especially for those such as KHS-Cnd, which counteract biofilms in vitro, but at rather high concentrations [29].

## Figures and Tables

**Figure 1 ijms-23-13494-f001:**
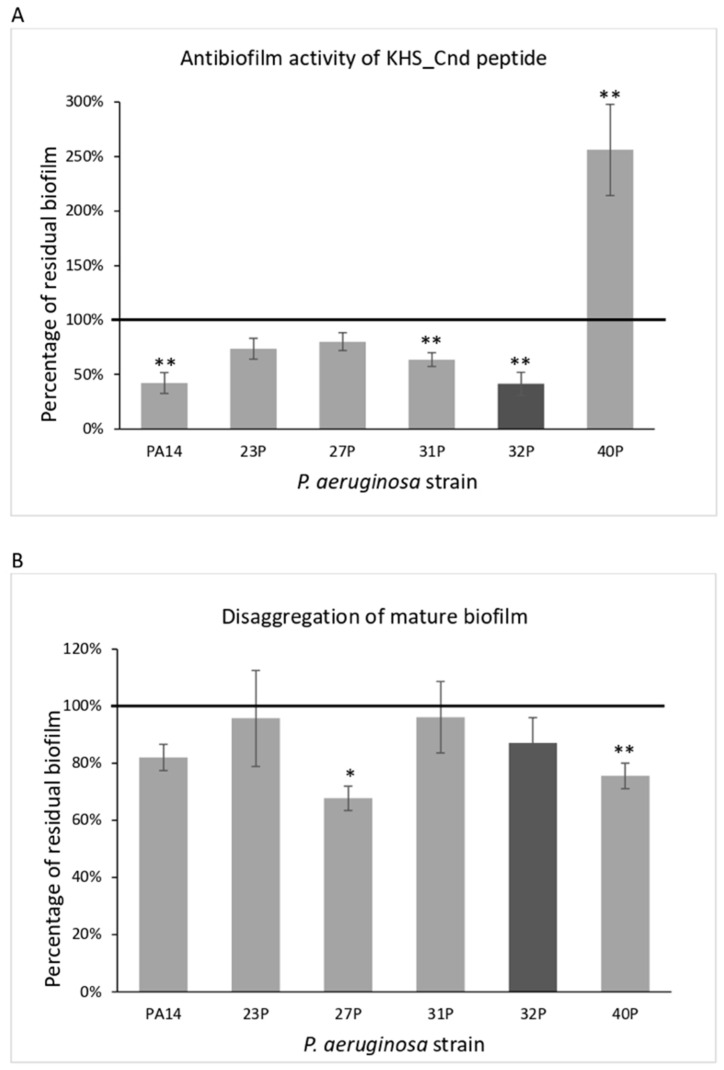
Effect of KHS-Cnd on biofilm formation by different clinical strains and the PA14 reference strain. Panel (**A**): Effect of KHS-Cnd on biofilm formation. KHS-Cnd was added to the culture medium at time zero (0 h, pre-adhesion period) and the biofilm was analyzed after overnight incubation. In the ordinate axis, the percentage of bacterial biofilm production is reported. Data are expressed as the percentage of biofilm formed in the presence of 40 µM KHS-Cnd (except for 32P where 20 µM KHS-Cnd was used) compared with the untreated bacteria. Each data point is composed of three independent experiments, each performed at least in three replicates. Panel (**B**): Effect of KHS-Cnd on the mature biofilm. KHS-Cnd was added to the culture medium after 24 h of biofilm growth (24 h of bacterial culture) and the biofilm was analyzed after overnight incubation. In the ordinate axis, the percentage of residual biofilm is reported. Data are expressed as the percentage of residual biofilm after 24 h of treatment with 40 µM KHS-Cnd (except for 32P where 20 µM KHS-Cnd was used) compared with the control sample. Error bars indicate the standard deviations of all of the measurements. Statistical difference was determined by the Student’s *t*-test: * *p* < 0.05; ** *p* < 0.01, compared with the control.

**Figure 2 ijms-23-13494-f002:**
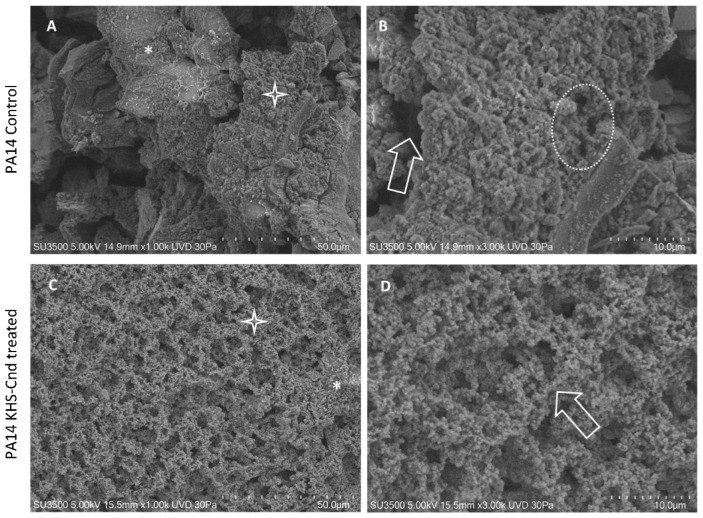
VP-SEM images of untreated *P. aeruginosa* PA14 (**A**,**B**). (**A**) Low magnification (1.00 K) showed smooth and compact ECM areas (asterisk), spongy, and rough areas (star). (**B**) At higher magnification (3.00K), ECM showed large meshes in some areas (arrow) and smaller ones in others (dotted oval). VP-SEM of *P. aeruginosa* PA14 treated with KHS-Cnd (**C**,**D**). (**C**) At low magnification (1.00 K), ECM appeared almost totally with a spongy appearance (star), and rare compact areas (asterisk) were present. (**D**) Higher magnification (3.00 K) showed that a 3D network of ECM trabeculae was formed by the aggregates of fine granular filaments having a “pearl necklace” aspect (arrow).

**Figure 3 ijms-23-13494-f003:**
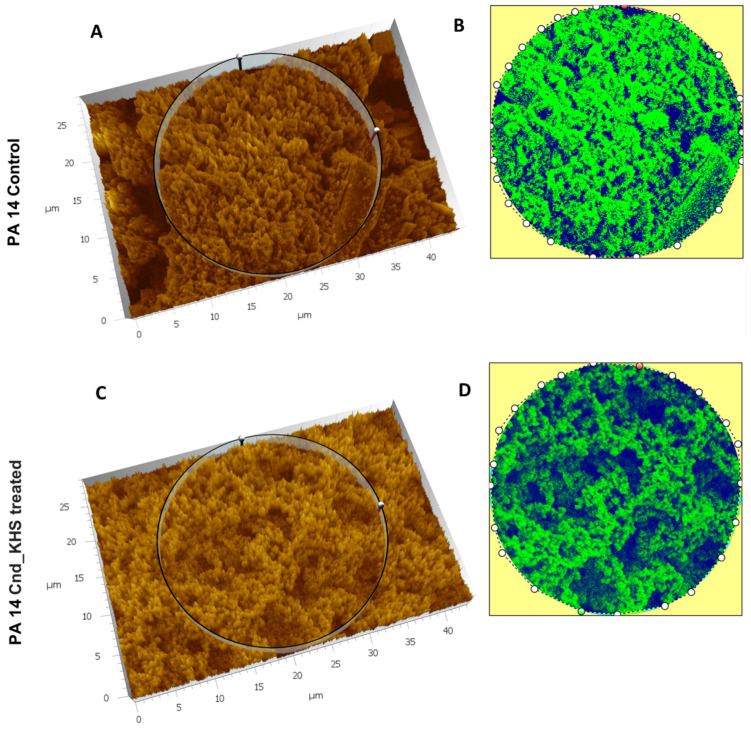
Three-dimensional reconstruction and analysis of *P. aeruginosa* PA14 (**A**,**B**) and *P. aeruginosa* PA14 treated with KHS-Cnd (**C**,**D**). The magnification is 3.00 K. (**A**) The spongy surface of untreated *P. aeruginosa* PA14 biofilm is imagined in 3D. (**B**) The circle represents the extracted area shown in false colors and related to untreated PA14. (**C**) The spongy surface of KHS-Cnd treated *P. aeruginosa* PA14 biofilm is imagined in 3D. (**D**) The circle represents the extracted area shown in false colors and related to KHS-Cnd treated *P. aeruginosa* PA14. High elevated areas corresponding to ECM trabeculae are colored in green. The labyrinthic channel system is represented in blue.

**Figure 4 ijms-23-13494-f004:**
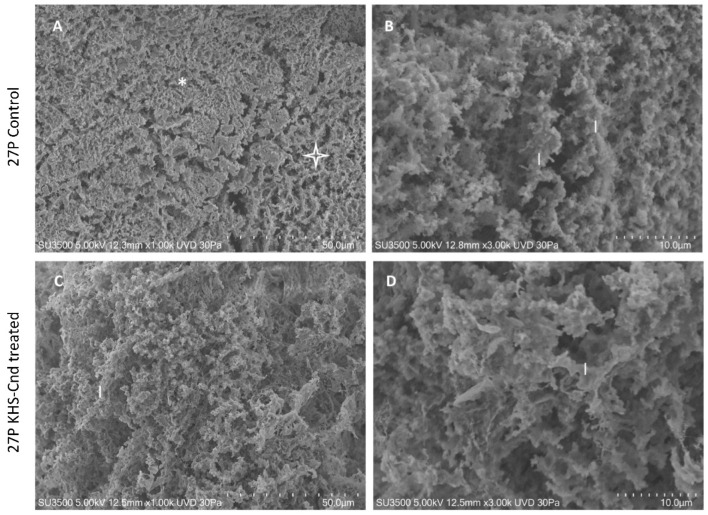
VP-SEM images of untreated *P. aeruginosa* 27P (**A**,**B**). (**A**) At low magnification (1.00 K), ECM was organized in both compact (asterisk) and spongy areas (star). (**B**) At higher magnification (3.00 K), ECM was formed by interconnected laminae (l), rather than trabeculae. VP-SEM images of *P. aeruginosa* 27P treated with KHS-Cnd (**C**,**D**). (**C**) At low magnification (1.00 K), the laminae forming ECM 3D network (l) appeared curly, as in the control sample. (**D**) Higher magnification (3.00 K) showed that the laminae (l) had the same shape as in the untreated sample.

**Figure 5 ijms-23-13494-f005:**
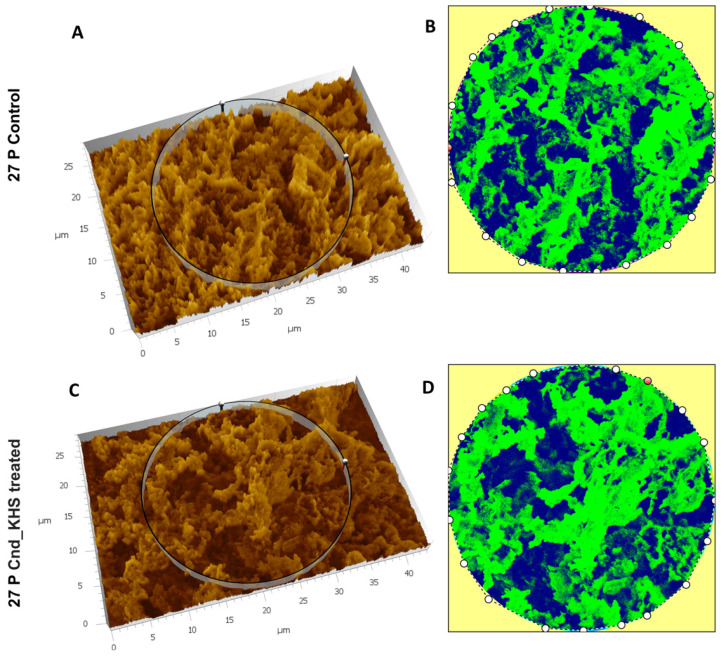
Three-dimensional reconstruction and analysis of *P. aeruginosa* 27P (**A**,**B**) and *P. aeruginosa* 27P treated with KHS-Cnd (**C**,**D**). The magnification is 3.00 K. (**A**) The spongy surface of untreated *P. aeruginosa* 27P biofilm was imagined in 3D. (**B**) The circle represents the extracted area shown in false colors and related to untreated 27P. (**C**) The spongy surface of KHS-Cnd treated *P. aeruginosa* 27P biofilm was imagined in 3D. (**D**) The circle represents the extracted area shown in false colors and related to KHS-Cnd treated 27P. High elevated areas corresponded to the edges of the laminae of ECM meshes and are colored in green. The labyrinthic channel system is represented in blue.

**Figure 6 ijms-23-13494-f006:**
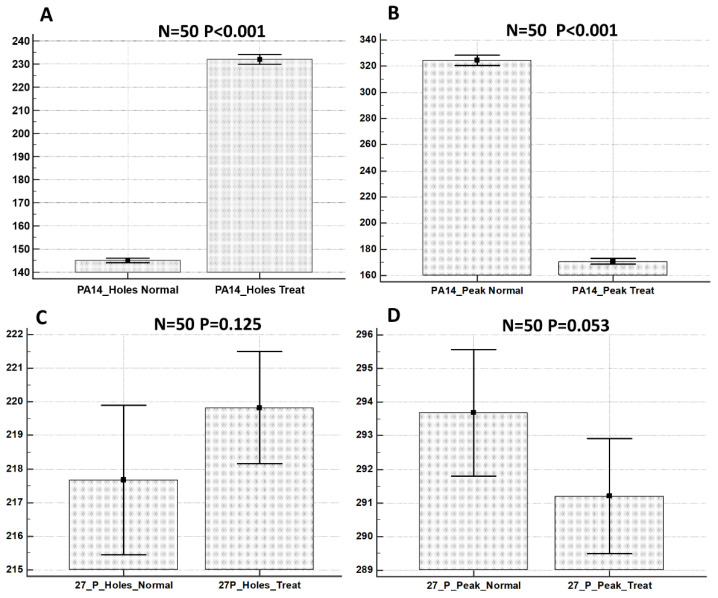
Statistical analysis of projected areas related to holes and peaks from: *P. aeruginosa* PA14 control sample (**A**); *P. aeruginosa* PA14 treated with KHS-Cnd (**B**); *P. aeruginosa* 27P control sample (**C**); and *P. aeruginosa* 27P treated with KHS-Cnd (**D**). N = 50 measurements were taken for each sample. Statistical difference was determined by the software MedCalc©: *p* < 0.05 was considered statistically significant.

**Figure 7 ijms-23-13494-f007:**
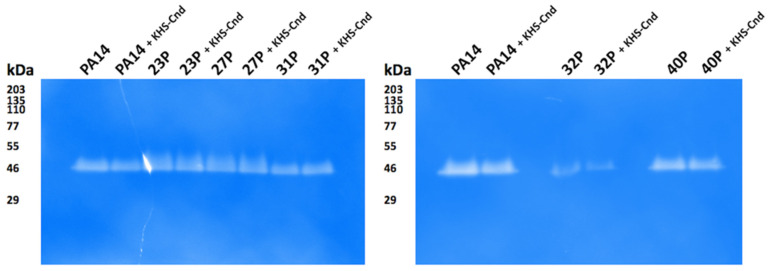
Gelatin-zymography to detect proteases secreted by clinical isolates of *P. aeruginosa*. Samples of unconcentrated culture supernatants were subjected to SDS-PAGE using 10% gels containing 0.2% gelatin. After gel incubation in renaturation buffer and Coomassie staining, clear bands in the zymograms corresponded to active proteases. The apparent molecular masses of protein standards are indicated in kDa.

**Figure 8 ijms-23-13494-f008:**
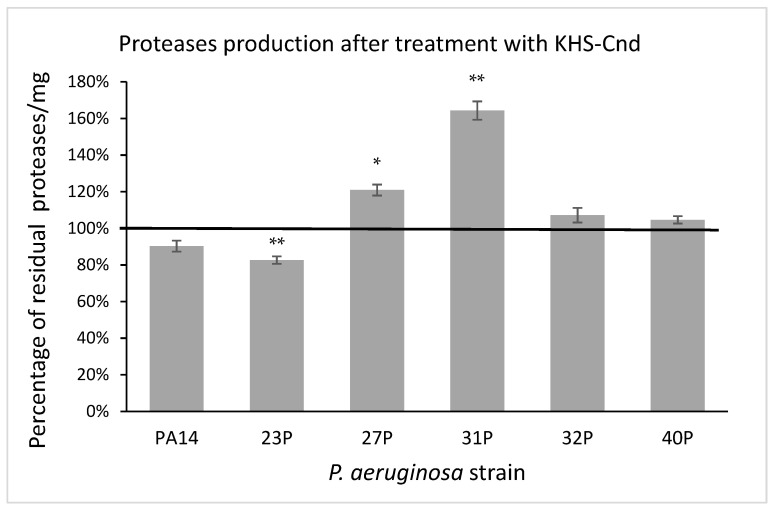
Effect of KHS-Cnd on the protease production of different clinical strains and PA14 reference strain. Data are expressed as the percentage of residual proteolytic activity after 24 h treatment with 40 µM KHS-Cnd (except for 32P where 20 µM KHS-Cnd was used) compared with the untreated control bacteria. Error bars indicated the standard deviations of all of the measurements. Statistical difference was determined by the Student’s *t*-test: * *p* < 0.05; ** *p* < 0.01 compared with the control.

**Figure 9 ijms-23-13494-f009:**
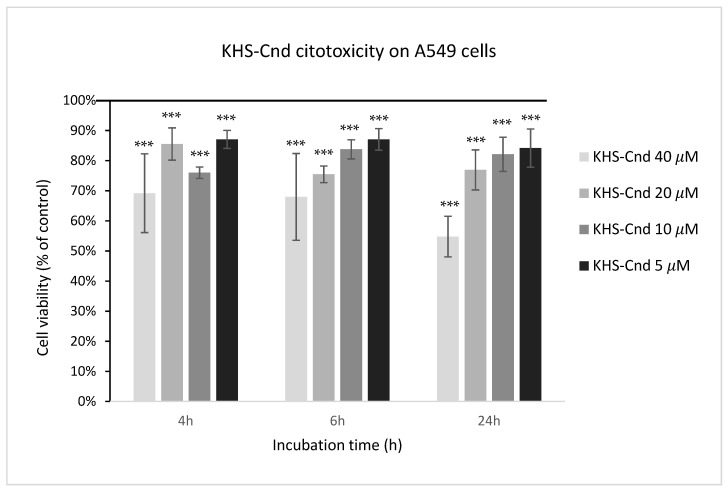
Effect of KHS-Cnd on the A549 immortalized cell line. Dose–response plot of cells after 4, 6, and 24 h of incubation with increasing concentrations (5–40 µM) of peptide. Cell viability was assessed by the 3-(4,5-dimethylthiazol-2-yl)-2,5-diphenyl-2H-tetrazolium bromide (MTT) assay and expressed as described in the Materials and Methods section. Values are given as the means ± SD (*n* = 6); *** indicates *p* < 0.001, with respect to the control cells.

**Table 1 ijms-23-13494-t001:** Phenotypic characterization of clinical and PA14 strains.

Bacterial Strain	Colony Phenotype	Protease Activity (OD 400 nm)	Biofilm 24 h ^a^ (OD 590 nm)	Biofilm 48 h ^b^ (OD 590 nm)
PA14	smooth	2.037 ± 0.020	3.561 ± 0.357	13.470 ±1.403
23P	smooth	1.988 ± 0.057	3.175 ± 0.851	0.738 ± 0.373
27P	smooth	2.175 ± 0.036	1.429 ± 0.643	3.049 ± 0.796
31P	mucoid	1.915 ± 0.001	1.741 ± 0.154	5.133 ± 0.946
32P	smooth	2.439 ± 0.078	1.117 ± 0.163	5.597 ± 1.390
40P	irregular colony edges	2.064 ± 0.058	0.970 ± 0.201	2.172 ± 0.194

^a^ Biofilm production during an incubation period of 24 h without medium replacement. ^b^ Biofilm production during an incubation period of 48 h with medium replacement after 24 h.

**Table 2 ijms-23-13494-t002:** The adhesion and invasion capabilities of *P. aeruginosa* on A549 cells in the presence and absence of 40 µM KHS-Cnd.

	Untreated		KHS-Cnd Treated	
	Adhesion	Invasion	Adhesion	Invasion
*P. aeruginosa* 27P	3.02 × 10^5^ ± 0.21 × 10^5^	8.06 × 10^3^ ± 0.08 × 10^3^	2.53 × 10^4^ ± 0.45 × 10^4^	3.0 × 10^3^ ± 0.2 × 10^3^

Data represent the mean ± SD of three independent experiments.

## Data Availability

Not applicable.

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
