# Peer review of "Anti-Virulence Potential of a Chionodracine-Derived Peptide against Multidrug-Resistant *Pseudomonas aeruginosa* Clinical Isolates from Cystic Fibrosis Patients"

_ijms, 2022, doi:10.3390/ijms232113494_

Round 1
Reviewer 1 Report
"Antivirulence potential of a chionodracine-derived peptide 2 against multidrug-resistant Pseudomonas aeruginosa clinical 3 isolates from cystic fibrosis patients" is an interesting study by Artini and colleagues regarding a novel peptide that breaks down biofilms produced by Pseudomonas isolates from CF patients. There are several concerns that need to be addressed prior to consideration for publication:
1). The rationale for choice of this peptide other than " isolated from the fish Chionodraco hamatus [20], as Antarctic marine 94 organisms are excellent candidates for discovering novel compounds due to the peculiar 95 environment they need to cope with" is not adequate. It is a difficult thought process to follow. If the authors have experience with forms of this peptide before, it would be entirely reasonable to refer to that experience.
2). In Figure 1, panel A, what time-point post-treatment with the peptide was the biofilm assessed? It is not clear from the figure legend. Also, in panel A P40 biofilm production is enhanced over its control, while in Panel B it is significantly diminished from control. This discrepancy in data needs to be explained or at least discussed.
3). I disagree with the authors conclusions regarding the effect of the peptide on biofilm architecture of 27P. Although the effect is not as robust as PA14, there is at least a strong trend over change (p=0.053).
4). I disagree with the authors conclusion on cell viability "As shown in Fig. 9, a slight toxicity was 318 observed for the higher tested concentration". The cell toxicity is seen at all doses from 5 to 40uM.
5). The tested dose of the peptide is fairly high as the authors acknowledge themselves as well. In patients with CF or COPD there is thick mucus production by the goblet cells lining the tracheobronchial tree in addition to the biofilm produced by Pseudomonas, a "double whamy" so to speak. Would such high local peptide concentrations be even achievable in a clinically meaningful way? I don't think that is likely, hence limiting the clinical utility of this approach.
6). There are multiple grammatical mistakes in the manuscript which at times make it hard to follow the thought processes. Please rectify these.
Reviewer 2 Report
I sugest tht authors discuss mre te fact of the amount of the KHS-Cnd peptide can present ssome toxicity at higher concentrations tested ( line 319). .What would be the alternatives to avoid toxicity. Use of a conjugates drug as mentioned in line 450?
Reviewer 3 Report
The article submitted to IJMS by Marco Artini et al., entitled “Antivirulence potential of a chionodracine-derived peptide 2 against multidrug-resistant Pseudomonas aeruginosa clinical 3 isolates from cystic fibrosis patients”, reports the data detected evaluating the antivirulence activity of a chionodracine-derived peptide (KHS-Cnd) against six Pseudomonas aeruginosa strains, one of which (strain P14) used as a reference strain, for its hyperbiofilm producer ability.
The experimental data are sound but the text is needs to be revised.
Major revision
Results
Please rewrite the section 2.1. It is quite incomprehensible and at times disconnected from figure 1.
- listing in advance the names of the 6 strains used. The current version of the text does not help immediately to understand how many strains there are, what they are and to create a link with figure 1. The number of tested strains is not reported in the title, abstract and introduction.
- Insert in this section a table with the characteristics of the 6 strains used. Alternatively, transfer Table 2 from Materials and Methods to Results. I believe it is appropriate.
- Concerning the figure 1, line 141, it says “Effect of KHS-Cnd on biofilm formation of different clinical and reference strains”. In the text and among the results only one reference strain is cited, the PA14. Please also review this part in all the figures as during the reading it creates discrepancies in the understanding of the above. Same in Figure 8.
- Please add in figure 1A Antibiofilm activity of KHS-Cnd peptide (0 h, pre-adhesion period) and in figure 2B Antibiofilm activity of KHS-Cnd peptide (24 h of bacterial culture). If you compare different time course, please indicate the times as reported in the text.
- Whay in figure 1A the strain 32P in bold/black, and not in figure 1B. Please revise.
- Line 145, figure 1 transfer “Each data point is composed of 3 independent experiments, each performed at least in 3-replicates” in materials and methods. Do it for any figures where is has been added (e.g. Figure 8).
- Line 114-115. In my opinion, if you use 0h as a starting time, you should use time zero in figure 1A and 24h in 1B. Please also revise this aspect.
Section 2.2
Please explain well why are selected the only strains PA14 and 27P. The Others?
Discussion
In line 418-419. very briefly is reported: “molecular mechanisms still remain to be clarified [31]”
It has been hastily written that little is known about the molecular mechanism by which these drugs act. It would be appropriate to expand this part, also describing the little known. A reader only understands that molecules would interfere with biofilm formation (0h) without understanding the reasons. It seems very simplistic to me.
The data does not reveal a homogeneous picture in the behavior of strains. There will probably be genetic factors that differentiate them and that underlie the different responses. Does the extracellular matrix structure have the same composition in all six strains? Is there any information? What kind of interaction is there between this type of peptides and biofilms? Also in prokaryotic species other than Pseudomonas aeruginosa.
Please, revise the text in general before to resubmission.
Minor revision
Line 41. Please report the list of prokaryotic species for which the acronym ESKAPE was established.
Line 52. Substitute the terms aggressive with altered.
Line 68. in CF patients
Line 331: Figure 9, write the acronym MTT in full.
Line 358-359. Please remove the sentence “Adhesion is expressed as CFU of bacteria that adhered to A549 cells 1 h post-infection at 37 °C. Invasion efficiency is expressed as 358 CFU of bacteria that were gentamicin resistant 1 h post-infection”. You already add in the text. As alternative, please transfer to materials and methods.
Line 365. Correct: … the hostile CF lung environment in … the hostile lung environment of CF patients.
Please don’t make confusion of writing. CF having lung ! Are the CF patients with lung! Please where necessary correct such sentence.
Line 371. In the discussion, at least the first time put again in full ECM.
Line 380. AMP
Line 517. Write the acronym ECM in full.
Round 2
Reviewer 1 Report
The authors have made extensive changes to the manuscript and addressed almost all concerns. In Figure 9 where they claim "almost negligible" cytotoxicity, even at the lowest dose (5uM), it is at least statistically significantly less viability of cells compared to controls with p-values<0.001.
Reviewer 3 Report
The manuscript has been significantly improved. Few further minor requests follow.
Line 120, Table 1.
Remove: “Phenotypic characterization of clinical and PA14 strains”. Leave only: Table 1.
Line 118-131.
Please substitute the text with the following reviewed:
Preliminary experiments were performed to assess the effect of KHS-Cnd on planktonic growth of six P. aeruginosa strains, five clinical strains and the hyperbiofilm producer PA14 reference strain [23]. The phenotypic characterization of clinical and PA14 strains were summarized in table 1.
Detected results showed that the peptide did not affect bacterial viability up to a concentration of 40 µM, except for 32P strain where an antimicrobial activity was evidenced at the highest used concentration. Considering these results, biofilm experiments were performed by testing KHS-Cnd at a concentration of 40 µM for all strains and, only for 32P, at a 20 µM concentration.
